# Real-world effects of medications for chronic obstructive pulmonary disease: protocol for a UK population-based non-interventional cohort study with validation against randomised trial results

Kevin Wing,[1] Elizabeth Williamson,[2] James R Carpenter,[2] Lesley Wise,[1] Sebastian Schneeweiss,[3,4] Liam Smeeth,[1] Jennifer K Quint,[5] Ian Douglas[1]

## ABSTRACT

**Introduction** Chronic obstructive pulmonary disease (COPD) is a progressive disease affecting 3 million people in the UK, in which patients exhibit airflow obstruction that is not fully reversible. COPD treatment guidelines are largely informed by randomised controlled trial results, but it is unclear if these findings apply to large patient populations not studied in trials. Non-interventional studies could be used to study patient groups excluded from trials, but the use of these studies to estimate treatment effectiveness is in its infancy. In this study, we will use individual trial data to validate non-interventional methods for assessing COPD treatment effectiveness, before applying these methods to the analysis of treatment effectiveness within people excluded from, or under-represented in COPD trials.

**Methods and analysis** Using individual patient data from the landmark COPD Towards a Revolution in COPD Health (TORCH) trial and validated methods for detecting COPD and exacerbations in routinely collected primary care data, we will assemble a cohort in the UK Clinical Practice Research Datalink (selecting people between 1 January 2004 and 1 January 2017) with similar characteristics to TORCH participants and test whether non-interventional data can generate comparable results to trials, using cohort methodology with propensity score techniques to adjust for potential confounding. We will then use the methodological template we have developed to determine risks and benefits of COPD treatments in people excluded from TORCH. Outcomes are pneumonia, COPD exacerbation, mortality and time to treatment change. Groups to be studied include the elderly (>80 years), people with substantial comorbidity, people with and without underlying cardiovascular disease and people with mild COPD.

**Ethics and dissemination** Ethical approval has been granted by the London School of Hygiene & Tropical Medicine Ethics Committee (Ref: 11997). The study has been approved by the Independent Scientific Advisory Committee of the UK Medicines and Healthcare Products Regulatory Agency (protocol no. 17_114R). An application

### Strengths and limitations of this study

► Large cohort study.
► Use of validated methods for detecting chronic obstructive pulmonary disease (COPD) within the Clinical Practice Research Datatlink.
► Use of randomised controlled trial (RCT) individual patient data to assess ability of non-interventional methods to detect COPD treatment effects within an RCT-analagous population.
► Adherence to medication will need to be assessed based on proxy variables (eg, time covered by prescription).

to use the TORCH trial data made to clinicalstudydatarequest.com has been approved. In addition to scientific publications, dissemination methods will be developed based on discussions with patient groups with COPD.

## INTRODUCTION

### Background and rationale

Chronic obstructive pulmonary disease (COPD) affects 3 million people in the UK.[1] The most common cause is smoking, and patients exhibit airflow obstruction that is not fully reversible. The disease is progressive, with declining lung function and a worsening of symptoms. Most troublesome are acute exacerbations manifested as a sudden worsening of symptoms, for example, severe coughing, shortness of breath and chest congestion, requiring urgent treatment and possibly hospitalisation. While smoking cessation remains the most effective intervention, the rate of exacerbation can be reduced by regular medication such as combination of long-acting beta-agonists (LABA) and

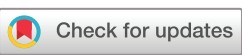

For numbered affiliations see end of article.

**Correspondence to**
Dr Kevin Wing;
kevin.wing@lshtm.ac.uk

inhaled corticosteroids (ICS) or long-acting muscarinic antagonists (LAMAs).[2 3]

COPD treatment guidelines are largely informed by randomised controlled trial (RCT) results,[4] but it is not clear if these findings apply to large patient populations not studied in trials. Fluticasone propionate+salmeterol (FP/SAL) is a LABA/ICS combination and is one of the most widely used COPD treatments. It was studied in large randomised trials (eg, the TORCH trial),[2] but the effects of treatment in important patient groups who were not studied are unknown. Some were excluded from trials (eg, those aged >80 years and those with substantial comorbidity) while others are under-represented (eg, people with mild COPD),[2 5] meaning conclusions about these groups are difficult to make.

While the conduct of non-interventional studies (sometimes also referred to as 'observational studies') to investigate possible drug harms is well established, the use of these studies to estimate treatment effectiveness is in its infancy. Issues of treatment channelling and indication bias mean that measuring the intended benefit of a treatment is beset with difficulties. Over the next few years, we will see more non-interventional studies of drug effectiveness emerging due to recent legislation requiring pharmaceutical companies to study the real-world effects of medications[6 7]; however, rigorous, validated methodology is needed to translate these complex data into reliable evidence. For example, the availability of anonymised individual patient data from RCTs provides the potential for 'RCT-analogous' cohorts to be selected from non-interventional data sources (by matching patient records from non-interventional data to the RCT patient records on key characteristics). If subsequent analysis of a non-interventional RCT-analogous cohort generates results that are similar to those generated by the reference RCT, one could be confident in the validity of the results, and in the non-interventional methods used to obtain these results in this setting.

In this study, we will use TORCH[2] individual trial data to validate non-interventional methods for assessing COPD treatment effectiveness, before going on to apply these methods to the analysis of treatment effectiveness within people excluded from or under-represented in the TORCH trial. Non-interventional data will be obtained from the UK Clinical Practice Research Datalink (linked to the Hospital Episodes Statistics database).[8] The results we generate will aid patients, prescribers and policy makers in deciding the most appropriate treatment for COPD for all types of patients. The approach used can also provide a template for treatment effectiveness research using non-interventional data with inbuilt validation against a randomised trial.

## AIMS AND OBJECTIVES

The aims of our study are (1) to measure the association between treatments for COPD and a number of COPD outcomes including exacerbation rate, mortality, pneumonia and time to treatment change among patients not included in randomised clinical trials for COPD treatments and (2) to develop a methodological framework with in-built validation against RCT data, for using non-interventional electronic health records (EHR) to answer questions about drug treatment effects (ie, both benefits and risks).

Specific objectives are to: (1) validate methods for measuring COPD medication effectiveness in EHR data by comparing with trial results; (2) use EHR data to measure COPD medication effectiveness in patients excluded from trials (most importantly those aged >80 years or with substantial comorbidity) and (3) determine COPD treatment effectiveness in an understudied disease stage (mild COPD).

## METHODS AND ANALYSIS

Figure 1 provides a high-level overview of the study, detailing each objective and data source used, and showing how existing RCT data will be used in objective 1 in order to validate methods for analysing COPD in routinely collected electronic health data that will then be applied to unanswered questions in objectives 2 and 3.

### Study design

We have chosen a cohort study design as it will allow us to measure the effects of prescribing different treatments for COPD on future outcomes in different types of patients. Eligibility criteria for cohort entry will vary between objectives (detailed in the 'Selection of participants' section).

### Setting/data sources

Patient data used in this study will be obtained from two different sources: the TORCH randomised trial and the UK Clinical Practices Research Datalink (CPRD) (linked to Hospital Episodes Statistics (HES) data).

### TORCH

TORCH was a placebo-controlled randomised trial of the combined inhaler FP/SAL for the treatment of COPD, published in 2007. Patients were randomised to receive FP/SAL, FP alone, SAL alone or placebo and the primary comparison of interest was between FP/SAL and placebo.[2] Key outcomes were expected benefits (rate of COPD exacerbation and mortality) and an expected harm due to the immunosuppressive action of the corticosteroid FP (pneumonia). While findings for the primary end point of mortality were null, this was thought to be due to poor statistical power as a result of a lower than anticipated mortality rate. Nonetheless, a lower rate of exacerbations was seen with FP/SAL, and a higher rate of pneumonia was observed. As one of the largest trials in COPD, and with 3-year follow-up, TORCH is a landmark study, providing a validation point for our study. We will obtain individual patient data from the TORCH study via www.clinicalstudydatarequest.com for use in objective 1 (see the 'Selection of participants' section).

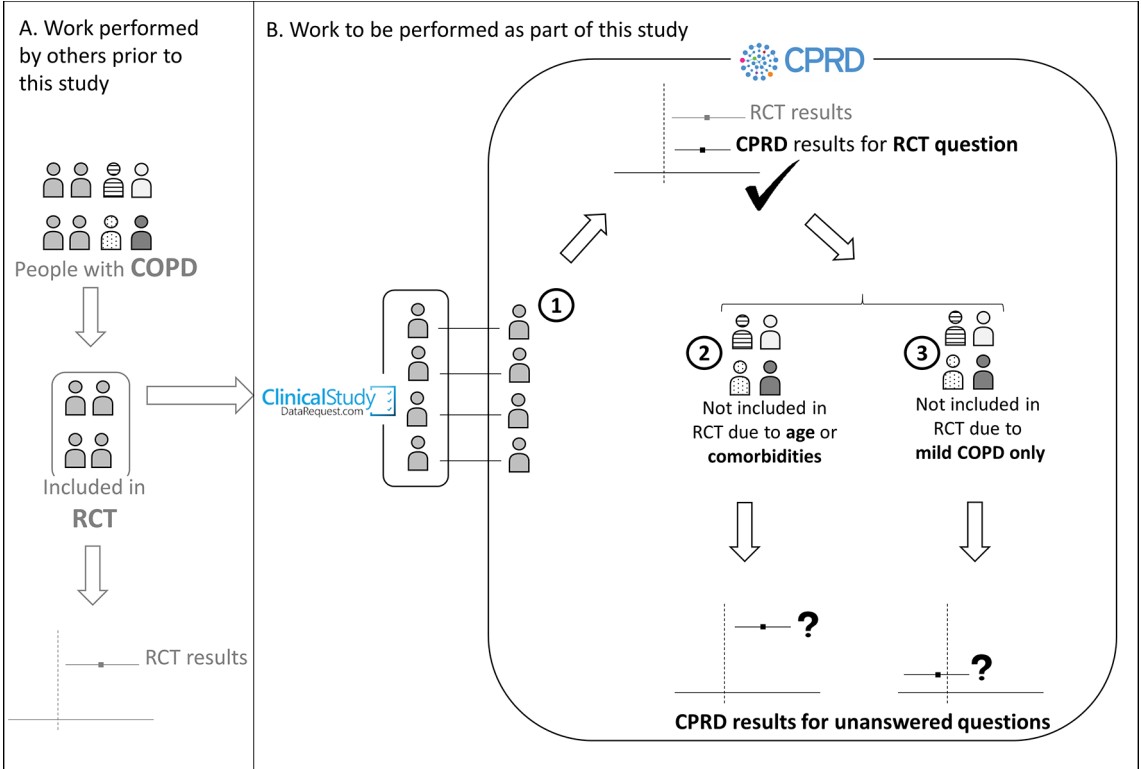

**Figure 1** Overview of study objectives and sources of data for the chronic obstructive pulmonary disease (COPD) real-world medicines effects study (RCT, randomised controlled trial, CPRD, the UK Clinical Practice Research Datalink, FP/SAL, fluticasone propionate+salmeterol). (A) Work performed by others prior to this study. Of the total population of people with COPD, only a subset are included in RCTs of COPD treatments, based on the RCT inclusion/exclusion criteria. The RCT generates results that inform clinical practice, and the anonymised raw data for the study can be made available to other researchers via the Clinical Study Data Request website. For this study, the specific COPD treatment RCT of interest is the TORCH trial,[2] investigating the effect of FP/SAL on COPD exacerbations. (B) Work to be performed as part of this study. (1) Objective 1: a cohort of TORCH (RCT)-analogous patients will be selected from the CPRD, by matching people with COPD within CPRD to the records of people included in the trial. An analysis of the effect of FP/SAL on COPD exacerbations will then be performed on this TORCH-analogous CPRD cohort. If the results obtained are comparable to those obtained in the TORCH trial itself, this will serve as a validation step, showing that data from the non-inverventional ('real-world') CPRD source can reliably be used to study COPD treatment effects. (2) Objective 2: the validated analysis techniques used for objective 1 will then be used to study people in CPRD who would not have been eligible for inclusion in an RCT due to their age and the presence of other comorbidities, and for whom the effect of FP/SAL is currently unknown. (3) Objective 3: the validated analysis techniques will then be used to study people with mild COPD only, who have been under-represented in RCTs, and for whom the effect of COPD treatments is unclear.

### UK Clinical Practices Research Datalink

The CPRD is a very large database of prospectively collected, anonymised UK population-based electronic health records. CPRD primary care records comprise ~8%–10% of the UK population and contain comprehensive information on clinical diagnoses, prescribing, referrals, tests and demographic/lifestyle factors.[8] In order to contribute to the database, general practices and other health centres must meet prespecified standards for research-quality data (ie, be 'up to standard'). Data quality/validity are therefore high and the data are nationally representative.[8 9] A patient starts contributing follow-up time to the database at the date they join an 'up-to-standard' practice (or the date that their practice starts contributing up-to-standard data), and stop contributing follow-up time on either their death date, their transfer out date (the date that

they leave the database due to reasons other than death) or on the last collection date for their practice. Linkage between the primary care records in CPRD and HES is well established for >60% of practices in the CPRD, providing a data set augmented with detailed secondary care diagnostic and procedural records. Algorithms have been established to identify COPD, COPD exacerbations and pneumonia (both hospital and primary care managed) in CPRD/HES-linked data (including validated algorithms for COPD and exacerbations).[10–12] See online supplementary material for a high-level overview of these algorithms.

### Selection of participants

Participants will be selected from the CPRD between 1 January 2004 and 1 January 2017. All patients will need to have been registered with an up-to-standard practice for

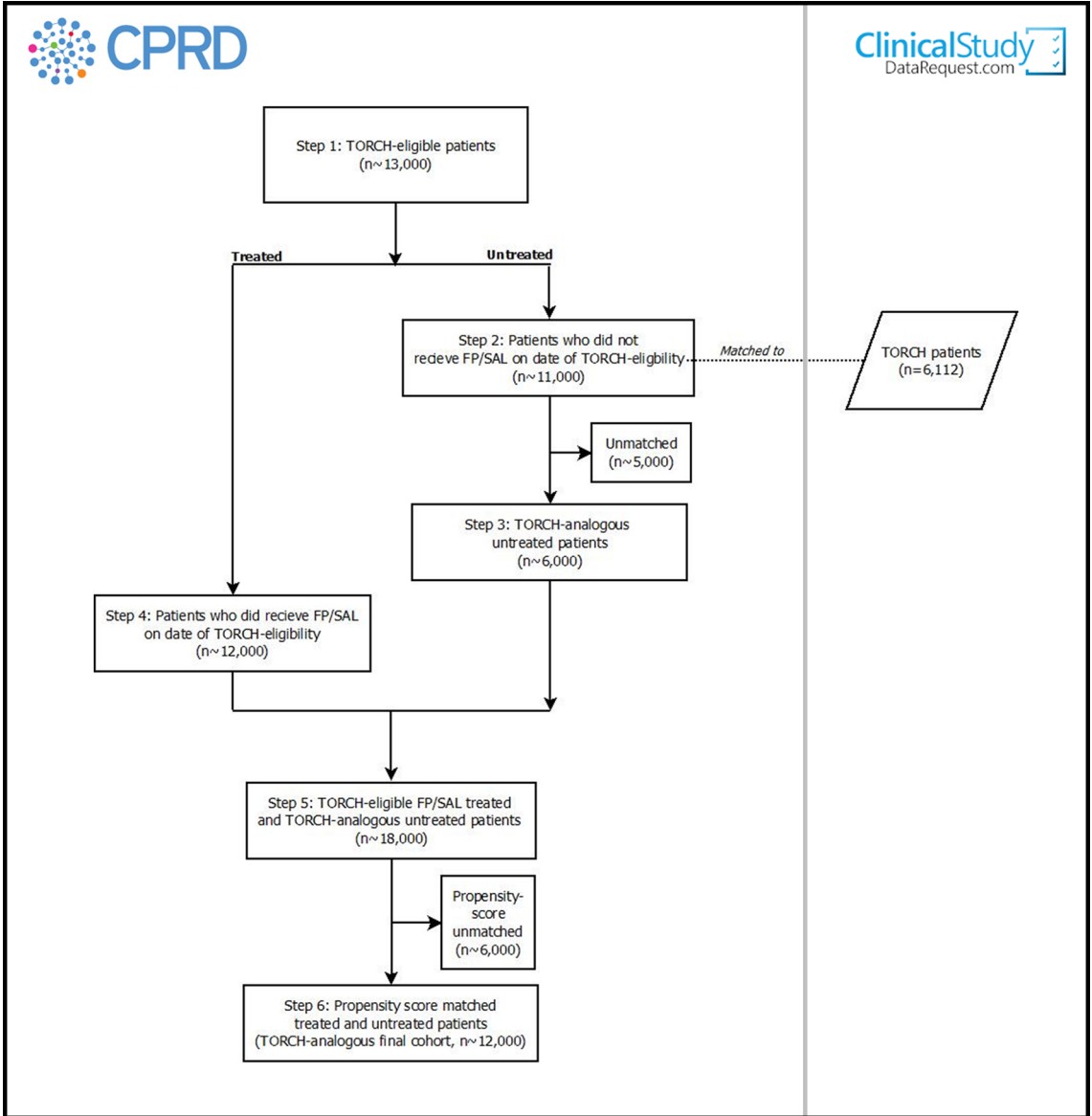

**Figure 2** Flow chart illustrating the planned selection of CPRD participants for objective 1 of the COPD real-world medicines effects study. Note in relation to step 5 (n~18 000) compared with step 1 (n~13 000): approximately 90% of the treated patients will also have been eligible as untreated patients, as they did not receive FP/SAL on their TORCH-eligibility date. This means that they will have at least one period of time during which they are untreated-eligible but then did subsequently go on to receive FP/SAL (meaning they have at least one period of time during which they are treated-eligible). If a person is included as both a treated and untreated participant, they will be contributing different periods of their person-time to each cohort (pre-FP/SAL treatment for the untreated vs post-FP/SAL treatment for treated), and this is handled in the analysis by assigning different index dates. COPD, chronic obstructive pulmonary disease; CPRD, UK Clinical Practice Research Datalink; FP/SAL, fluticasone propionate+salmeterol.

at least 12 months. Participant selection criteria will then vary by objective as detailed below.

### Objective 1: validation of non-interventional methods by comparing with trial results

An overview of each of the steps for participant selection for objective 1 is provided in figure 2.

#### Step 1

We will select all patients in the CPRD with COPD who are eligible for HES-linkage and during the period covered

by the linkage would have met the following TORCH study inclusion criteria:

► a diagnosis of COPD;
► age 40–80 years;
► smoking history;
► lung function (forced expiratory volume in 1 s ($FEV_1$) <60% predicted, $FEV_1$/forced vital capacity (FVC) ratio <70%).

An eligible-for-inclusion date will then be assigned as the date that all of the above inclusion criteria were met for the individual. We will then exclude any individual

who has any of the following TORCH study exclusion criteria prior to their eligible-for-inclusion date:

► a diagnosis of asthma (within the previous 5 years);
► a diagnosis for any (non-COPD) respiratory disorder;
► a record of lung surgery;
► a diagnosis of alpha-1 antitrypsin deficiency;
► evidence of drug or alcohol abuse;
► a record of having received long-term oxygen therapy;
► diagnoses for conditions likely to interfere with the TORCH trial or cause death within 3 years;
► current use of oral corticosteroid therapy (defined as continuous use for >6 weeks, with courses of oral corticosteroids separated by a period of <7 days considered as continuous use);
► any exposure to FP/SAL within the previous 4 weeks.

Finally, in-line with the TORCH trial approach, anyone who has an exacerbation requiring oral corticosteroid therapy or hospitalisation during the run-in period (the 2-week period following eligibility) will also be excluded.

Feasibility counts in the CPRD indicate there are ~13 000 patients meeting these criteria (figure 2). Given the limited information on how asthma exclusions were applied in the TORCH study, we will perform a sensitivity analysis in which the asthma exclusion is a diagnosis within the previous 1 year, rather than 5 years as specified above.

### Step 2

Next we will determine if/when these patients ever received FP/SAL. During any time between attaining TORCH eligibility and a subsequent prescription for FP/SAL, patients will be eligible for inclusion as an unexposed (to FP/SAL) patient in objective 1. There may be multiple time periods within a person's record where eligibility as an unexposed patient is met (figure 3). Feasibility work shows that between 1 January 2004 and 1 January 2017 there were ~11 000 TORCH-eligible patients in CPRD who did not receive FP/SAL at the time they attained TORCH eligibility and therefore have at least one time period that means they are eligible for inclusion as an objective 1 unexposed participant (figure 2). Individuals in CPRD who have more than one unexposed eligibility period within their record (figure 3) will be able to contribute more than once to the pool of unexposed participants (with the covariates and person-time contributed unique to the specific unexposed eligibility period).

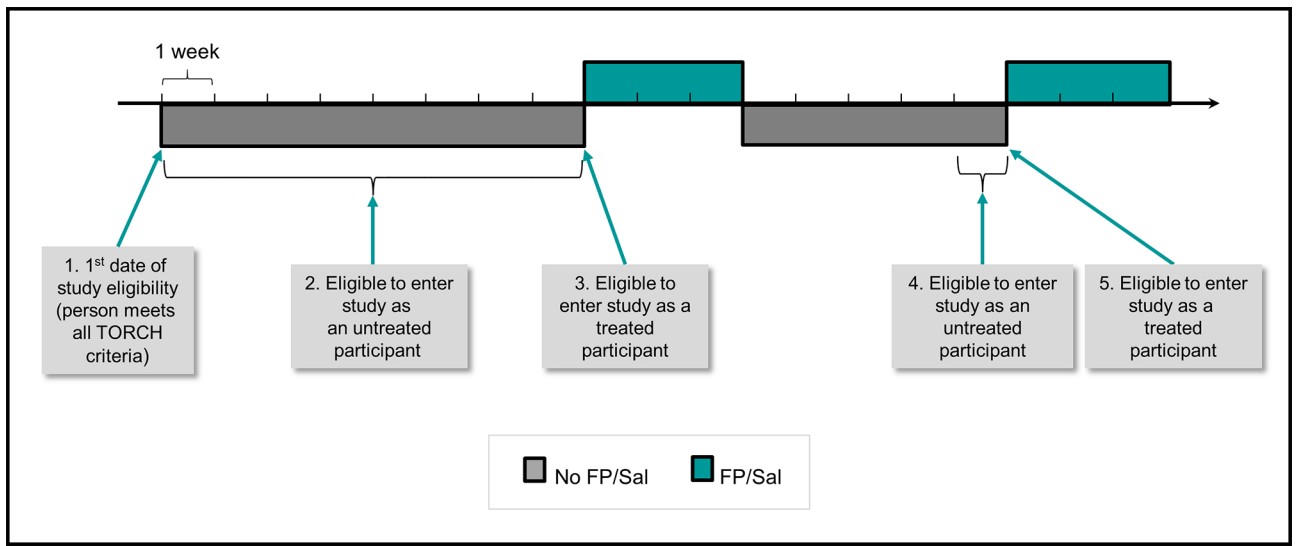

**Figure 3** Example timeline for a person in CPRD who is eligible for participation in objective 1, illustrating multiple periods of unexposed/exposed (FL/SAL untreated/treated) eligibility. (1) First date of study eligibility: person meets TORCH eligibility as detailed in objective 1, step 1 on this date that is, has a diagnosis of COPD, is between 0 and 80 years, $FEV_1$ <60% predicted and $FEV_1$/FVC ratio <70%, smoking history, no asthma history, no lung surgery history, no long-term $O_2$ therapy, no alpha-1 antitrypsin deficiency, no drug/alcohol abuse, no exposure to FP/SAL within the previous 4 weeks. (2) Eligible to enter study as an untreated participant: patient can be selected as an untreated participant on any date within this period (as detailed in step 2). (3) Eligible to enter study as a treated participant: FP/SAL treatment starts, patient is able to be selected as a treated study participant on the date that FP/SAL treatment starts (as detailed in step 4). (4) Eligible to enter study as an untreated participant: patient stops treatment, but is not immediately eligible for selection again as an untreated study participant. After 4 weeks of no FP/SAL treatment however, they meet the TORCH eligibility ciriteria, and may be selected at any date during the remaining one untreated week as an untreated patient. This is the second untreated period that this person can contribute to the total pool of untreated period records that will be available for matching to the TORCH participants (as detailed in steps 2 and 3). (5) Eligible to enter study as a treated participant: FP/SAL treatment (re)starts, patient can be selected as a treated study participant on the date that FP/SAL treatment (re)starts. This is the second treated period that this person can contribute to the total pool of treated period records that will be available for propensity score matching to the untreated participants (as detailed in steps 4–6). COPD, chronic obstructive pulmonary disease; CPRD, UK Clinical Practice Research Datalink; $FEV_1$, forced expiratory volume in 1 s; FP/ SAL, fluticasone propionate+salmeterol; FVC, forced vital capacity.

### Step 3

Having obtained individual-level patient data for TORCH participants from clinicalstudydatarequest.com, we will then match each TORCH participant (n=6112) 1:1 with the closest available unexposed patient record in the CPRD pool of FP/SAL untreated patients obtained in step 2. Matching will be based on the following TORCH baseline characteristics:

► age
► sex
► body mass index
► previous treatment with:
  – inhaled corticosteroids
  – LABA
► history of COPD exacerbations
► history of cardiovascular disease
► lung function.

Where an individual from CPRD has multiple unexposed 'eligibility periods' that can be matched to a TORCH participant, the CPRD characteristics that will be matched on will be those from the beginning of the specific eligibility period.

Some of the TORCH inclusion criteria will not be fully assessable using CPRD data (eg, we will be able to assess whether patients are smokers but will not always know their pack year history). Hence, the inclusion/exclusion criteria are analogous with TORCH criteria but we acknowledge they are not identical. Identification of criteria will be done based on algorithms already determined and by the identification of clinical codes in the CPRD. For those individuals that have contributed multiple unexposed records to the pool of CPRD unexposed participants (figure 3), after one of their unexposed records has been matched to a TORCH participant we will remove all of their other unexposed records (meaning that an individual can only appear once in the final TORCH-matched unexposed cohort). We anticipate matching all or the majority of TORCH participants with a CPRD patient, giving us a pool of TORCH-analogous untreated patients within CPRD, with similar baseline characteristics as TORCH participants at the point of randomisation (n~6000, figure 2).

### Step 4

Following this, we will select all patients in CPRD meeting the TORCH eligibility criteria specified above, and who also received treatment with FP/SAL (either on the date of eligibility or at a later date). From feasibility work, we anticipate ~12 000 eligible FP/SAL-treated patients (figure 2), some of whom may have multiple time periods of treated patient eligibility and therefore could contribute more than once to the initial pool of treated patients (figure 3). In contrast to the unexposed TORCH-eligible cohort, our initial approach will not involve matching participants of the exposed (to FP/SAL) TORCH-eligible cohort with participants from the TORCH trial, as this would negatively impact the ability to calculate propensity scores for receiving FP/SAL in step 5. Note that there

will be overlap between those selected as untreated (in step 2) and as treated (in step 4). Approximately 90% of the step 4 treated patients will also have been eligible as step 2 untreated patients, as they will have had periods where they were not treated with FP/SAL and met the step 2 untreated eligibility criteria in addition to separate periods where they were treated with FP/SAL and met the step 4 treated eligibility criteria (figure 3). If a person is included in both the untreated step 2 and treated step 4 cohorts, they will be contributing different periods of their person-time to each cohort (pre-FP/SAL treatment for step 2 vs post-FP/SAL treatment for step 4), and this will be handled in the analysis by assigning different index dates for step 2 compared with step 4.

### Step 5

We will combine the CPRD groups obtained in steps 3 and 4 (n~18 000, see note in step 4 relating to 90% overlap) and using their baseline characteristics will calculate propensity scores for receiving FP/SAL. The propensity score calculation will be based on a wide range of covariates (see the 'Statistical analysis' section for full details). Where a participant is contributing more than one treated time period record to the pool of exposed records (as described in step 4), baseline characteristics will be updated at the beginning of each treatment-eligible period. Multiple eligible treatment periods from a single person are then included in the propensity score model as if they came from different individuals. The variables selected for the score will then become the basis for propensity score modelling in objectives 2 and 3.

### Step 6

Each untreated patient derived in step 3 (n~6000) will be matched 1:1 with the FP/SAL-treated patient record from step 4 with the closest propensity score (~12 000) giving us an analysis population for objective 1 of ~12 000 patients—double the size of TORCH (figure 2). For those individuals that have contributed multiple exposed records to the pool of CPRD-exposed participants (figure 3), after one of their exposed records has been propensity-score matched to an unexposed participant, we will remove all other exposed records for that individual from the remaining pool of CPRD-exposed participants. This will mean that an individual can only appear once as an exposed participant in the final propensity-score matched cohort.

We will also apply an alternative additional approach for objective 1, where instead of generating and using propensity scores to obtain a final analysis population at steps 5 and 6, we will match records from our exposed TORCH-eligible cohort with participants from the TORCH trial to create a TORCH-analogous exposed patient cohort. This will then be combined with the TORCH-analogous unexposed patient cohort to create a final analysis population (with multivariable regression techniques used to account for confounding instead of propensity scores).

## Objective 2: measurement of COPD treatment effects in patients excluded from trials

We will select separate cohorts of patients who have a valid COPD diagnosis in the CPRD[8] and who would not have been eligible for inclusion in the TORCH trial (and therefore also not eligible for our objective 1) due to the following characteristics: (1) age >80 years OR (2) history of lung surgery OR (3) history of long-term oxygen therapy OR (4) evidence of drug/alcohol abuse OR (4) an asthma diagnosis at any time prior to inclusion OR (5) substantial comorbidity. In relation to substantial comorbidity, TORCH required people to be excluded from the trial if they had serious uncontrolled disease with a likelihood of causing death within 3 years. It is likely this criterion affected participant selection and led to a lower overall rate of death than originally anticipated, although we recognise this criterion is subjective. During objective 1, we will be able to select groups of people who were generally not included despite being eligible, most likely because of this subjective exclusion criterion. We anticipate this will be people with substantial comorbidity, for example, serious vascular disease. Status for such diseases is readily identified in both the CPRD data and in the TORCH baseline data. We will only be able to specify this criterion in detail after we have completed objective 1.

Participants for each of the objective 2 cohorts will be selected in a similar fashion to the objective 1 cohort, with the amended eligibility criteria specified above applied (ie, step 1 will be modified for selection of each of the objective 2 cohorts). As for objective 1, each participant will be allowed to have multiple FP/SAL exposed and unexposed eligibility periods in their record, as described in figure 3. In contrast to objective 1, there will be no matching of unexposed patients to TORCH patients, as we do not require a TORCH-analagous cohort for this analysis (ie, no step 3). All other selection steps will be as applied for objective 1, including the use of propensity score matching in order to obtain comparable unexposed and exposed groups for analysis.

## Objective 3: determination of treatment effects in an understudied disease stage

We will select separate cohorts of patients who have a valid COPD diagnosis in the CPRD[8] and who would not have been eligible for inclusion in the TORCH trial (or our objective 1) due to the following characteristics: (1) >60% predicted $FEV_1$ (or >50% plus Medical Research Council breathlessness scale 1 or 2, or >50% plus COPD Assessment Test score <10) and (2) no exacerbations in the year post-COPD diagnosis. We will also perform a sensitivity analysis where we allow the group of people with $FEV_1$ >60% predicted who had a maximum of one exacerbation within 1 year post-COPD diagnosis to be included. As for objective 2, the selection steps will be similar to objective 1, with modified criteria for step 1 and the removal of the TORCH-matching step (step 3).

## Exposures, outcomes and covariates
### Exposures

For all objectives, exposures will be determined using CPRD prescribing records and code lists for COPD treatments (codelists provided in online supplementary material).

For objective 1, use of FP/SAL (trade name Seretide) is the primary exposure of interest and will be compared with no treatment with FP/SAL. We will limit included patients to those receiving Seretide 500/50, the dose used in TORCH. This information is recorded for all prescriptions of Seretide and this dose is the only currently approved dose for COPD in the UK (although we recognise some prescribing may not follow the licensed indication). If the results for our FP/SAL versus no treatment comparison are not consistent with the TORCH FP/SAL versus placebo results (see the 'Statistical analysis' and 'Validation of results against TORCH' sections for a definition of consistent), we will perform additional analyses where instead of using a no-treatment comparator group, our objective 1 comparator group will be people exposed to SAL, one of the other comparator groups from the TORCH trial.

As a secondary analysis in objective 1, other treatments for COPD will also be compared with no treatment. Selection of unexposed and exposed people for each of these drugs will follow steps 1–6 detailed above in the 'Selection of participants—objective 1' section, although step 3 will be omitted (as these cohorts will not need to be matched to TORCH). The other treatments we plan to include are as follows:
a. LABA
b. LAMA
c. LABA+LAMA
d. LABA+ICS other than FP/SAL
e. LABA+LAMA+ICS.

For objectives 2 and 3, we will again use recorded prescribing information to determine the dose received. We will be reliant mostly on the strength of each individual drug, which is recorded automatically against each product and does not require general practitioners (GPs) to enter these data, ensuring completeness. We will then be able to stratify analyses based on the dose prescribed. Specific exposures for objectives 2 and 3 are as follows (all vs no treatment):
a. FP/SAL
b. LABA
c. LAMA
d. LABA+LAMA
e. LABA+ICS
f. LABA+LAMA+ICS.

### Outcomes

Outcomes to be measured are as follows:
1. COPD exacerbation: to be defined using a CPRD-HES algorithm developed previously by authors of this study protocol.[10]

2. All-cause mortality: as recorded in Office of National Statistics (ONS) mortality statistics (data that is linked to CPRD data).

3. Pneumonia: as defined using a CPRD-HES algorithm published previously by authors of this study protocol.[11]

4. Time to COPD treatment change: determined by prescribing records indicating the start of a new, additional COPD treatment.

### Covariates

Covariates to be considered for inclusion in the propensity score include the following (all obtained from CPRD data):

► Lung function ($FEV_1$, $FEV_1/FVC$)
► Age
► Gender
► Body mass index
► Alcohol consumption
► Vascular disease (broken into individual components, eg, hypertension, heart failure, atherosclerotic disease)
► Use of prescribed aspirin and statins
► Prior treatment with other COPD medication
► Type 2 diabetes
► History of cancer
► Renal disease
► Healthcare utilisation intensity (number of prior visits, hospitalisations, number of distinct medications used, number of procedures).

### Sample size
#### Objective 1

Assuming a baseline conservative exacerbation rate of 0.5 per patient per year,[10] we would only require a sample of 408 patients per treatment group to detect a reduction in annual exacerbation rate to 0.4 per year, with 80% power and 5% significance. The estimated sample size is ~12 000, which will provide ample power for the main outcomes of interest, and allow stratification by patient characteristics to determine stratified results, and will also be ample for the secondary analyses where we will use 99% CIs. For example, to detect a reduction from 0.5 to 0.4 exacerbations per year with 80% power and 1% significance we would need ~600 people in each treatment group.

#### Objectives 2 and 3

We are also confident that we will have sufficient numbers to allow well-powered analyses for objectives 2 and 3. For example, a feasibility count looking at the number of people over the age of 80 eligible for inclusion in objective 2 estimated that there would be >2000 people in each exposure group.

### Statistical analysis
#### Propensity score for addressing confounding

The propensity score will be constructed using the principle that predictors of the exposure and outcome, or outcome only (mortality) should be included. We will consider a wide range of factors for inclusion (as listed in the 'Covariates' section), such as: age, sex, body mass index, alcohol consumption and a wide range of comorbidities (eg, type 2 diabetes, coronary heart disease, cerebrovascular disease, peripheral vascular disease, heart failure, hypertension, renal disease, cancer). We will further adjust for healthcare utilisation intensity (number of prior visits, hospitalisations, number of distinct medications used, number of procedures, etc) as these are generic correlates of disease state and the likelihood of recording completeness. We have substantial prior experience of building propensity models.[13–16]

For the additional alternative approach to objective 1 relying on matching of both unexposed and exposed patients to the TORCH trial patients (described in the 'Selection of participants—objective 1—step 6' section), we will use multivariable regression techniques to address confounding, considering a similar wide range of covariates for adjustment.

The variable list used for the propensity score model obtained in objective 1 will be the basis for propensity score modelling in objectives 2 and 3, but additional variables will also be considered given the different nature of the patient populations being studied in these objectives.

### Methods of analysis

For all objectives, comparisons will be made according to FP/SAL (or other drugs being analysed as specified in the 'Exposures' section) status for rate of COPD exacerbation, pneumonia and mortality over 3 years. All analyses will be performed according to the 'intention-to-treat' principle (as was done in the TORCH study), meaning that if a participant enters the study as either an exposed or unexposed participant, they will remain assigned to that exposure category for the entire duration of their follow-up (irrespective as to whether their true exposure status changes). For exacerbations, a negative binomial model will be used, accounting for variability between patients in the number and frequency of exacerbations, with the number of exacerbations as the outcome and the log of treated time as an offset variable. Time to mortality, first pneumonia and treatment change will be analysed using Cox proportional hazards regression. This mirrors TORCH end points of major benefit and harm. We anticipate the propensity matching process will allow us to assemble treated and untreated groups that are very similar with respect to baseline characteristics except FP/SAL treatment status. However, this will be tested by assessing standardised differences for each baseline variable. If substantial differences are noted for important variables, it may be necessary to further adjust the statistical models. This could also include examining the effect of using greedy versus optimum matching approaches in order to obtain the closest propensity score match and/or matching at a ratio other than 1:1.[17]

## Validation of results against TORCH

We will validate our findings against TORCH as part of objective 1 by determining whether results of the CPRD FP/SAL versus no FP/SAL treatment analysis are compatible with the TORCH exacerbations rate ratio for FP/SAL versus placebo (0.75; 95% CI 0.69 to 0.81). This outcome has been selected as it is an outcome of key significance for people with COPD[4] and the result in TORCH shows a clear benefit with 95% confidence limits below 1. We have set two criteria that must be met for us to conclude results are consistent. First, the effect size must be clinically comparable with TORCH findings; the rate ratio for exacerbations in CPRD must be between 0.65 and 0.9. This range is deliberately not symmetrical around the TORCH estimate of 0.75 as we would anticipate the treatment effect in routine clinical care may be weaker than that seen in the optimised setting of a randomised trial. We recognise this rule could be met with a poorly powered, inconclusive result, so a second criterion is that the 95% CI for the rate ratio must exclude 1. If we go on to compare FP/SAL with SAL alone (see the 'Exposures, outcomes and covariates' section, 'Exposures' subheading), the 95% CI would also need to exclude 1, and the rate ratio would need to be between 0.81 and 0.95 (compared with the TORCH FP/SAL vs SAL result of 0.88, 95% CI 0.81 to 0.95).

## Handling measurement of adherence to medication

Adherence to issued prescribing in general practice is likely to vary according to the treatment issued, for example, short course antibiotic treatment is notoriously not well adhered to, whereas long-term life-saving treatment such as antiretroviral medication is more likely to be taken as prescribed. While we do not have figures for adherence for COPD medication in UK general practice, we are able to estimate the proportion of time covered by prescribing as a proxy for adherence and will account for this in our analyses. Moreover, our intention is to estimate the effect of prescribing at the population level, and to some extent, the clinical effects we will measure are in part due to pharmacological effects, and in part the way the treatment is taken which includes adherence. Also of note, prescribing for COPD in the UK is predominantly through GPs and so we will not be missing prescribing information from other potential sources of treatment.

The data analysis for adherence will necessarily be a significant element of the work to be done for this study. However, we have reviewed the records for a random sample of 30 people with COPD starting treatment with FP/SAL to look at adherence patterns over the course of a year. Of the 30 patients, 20 (67%) were still receiving Seretide (FP/SAL) 1 year after starting treatment. Of the 20 who received Seretide for a full year, 15 (75%) received sufficient prescriptions to suggest at least 50% adherence over the year and 8 (40%) had sufficient prescriptions to suggest 80% adherence or higher. As expected, this suggests two things: first, adherence is likely to be poorer in routine clinical care than in the trial population; in

TORCH 80% of participants were estimated to have adherence at 80% or higher. Second, there is a wide range of adherence in routine care. This will allow us to estimate both the population-level effects of treatment as actually used in routine care and to estimate the treatment effect in patients with more similar levels of adherence to TORCH participants. While we acknowledge that prescribing can only be a proxy for used medication, we believe it is not an unreasonable assumption that the amount of medication prescribed is correlated with the amount consumed. We plan to assess adherence for the cohort that we select for objective 1 beyond 1 year and report the findings. In the event that objective 1 detects a null or poorer treatment effect than anticipated (rate ratio >0.9), we will conduct a sensitivity analysis restricted to people estimated to be covered by FP/SAL treatment for 80% of their follow-up.

## Misclassification of drug exposure periods and outcome status

It is possible that an individual may still be exposed to FP/SAL for some time after a prescription has finished, for example, if they have medication at home that they have not used from a previous prescription. This would mean that people may become eligible for inclusion in the unexposed group while they are actually still exposed. If our result differs from the TORCH results (eg, a rate ratio <0.65 or >0.9), we will conduct a sensitivity analysis in which we include an additional (grace) exposed period equivalent to the length of a single prescription at the end of each actual exposed period, and only classify individuals as eligible for inclusion as unexposed at the end of this additional period.

Our results could also be impacted by misclassification of outcome, given the routine nature of the data. Our initial approach for detection of COPD exacerbations is to use a validated case definition from previous work that maximises positive predictive value while maintaining a relatively high sensitivity.[12] If our result differs from the TORCH results, we will consider performing a sensitivity analysis in which we assess the impact of applying alternative case definitions for COPD exacerbations (see online supplementary material for an overview of articles relating to the case definitions we plan to use, including any validity measurements provided).

## Missing data

CPRD data are shown to be almost complete for drug prescribing and mortality (partly through ONS linkage). Smoking history tends to be very well recorded for people with COPD and missingness is likely to be minimal.[10] Information on important comorbidity is also well recorded in CPRD. We will conduct both complete record analyses and use multiple imputation where appropriate assumptions hold, applying findings from methodological work led by one of the study team (EW) into the use of multiple imputation in propensity score modelling.[16]

## ETHICS AND DISSEMINATION
### Approval by ethics and scientific committees

An application for scientific approval related to use of the CPRD data has been approved by the Independent Scientific Advisory Committee of the Medicines and Healthcare Products Regulatory Agency (protocol no. 17_114R). CPRD data are already approved via a National Research Ethics Committee for purely non-interventional research of this type.

An application for use of the TORCH trial individual patient data was made to the clinicalstudydatarequest.com site, which is checked by the Wellcome Trust and relevant sponsors to make sure information is complete and that the sponsor's requirements for informed consent have been met. The application is then sent to an independent review panel that consider the scientific rationale, objectives, publication plan, conflicts of interest and qualifications and experience of the research team before making a decision on providing access to the data. We recently obtained approval of all aspects of this application.

### Dissemination plans

Dissemination of findings will be via a combination of channels. The work will be published in high ranking peer-reviewed journals and we anticipate three publications to arise directly from the planned work. Findings will also be presented at relevant scientific conferences such as the British Thoracic Society Conference and the European Respiratory Society International Congress. We will also engage with patients already identified from a clinic run by one of the authors of this protocol (JQ) and from Breathe Easy Groups and with relevant charities such as the British Lung Foundation to determine the most relevant ways to disseminate results directly to patients in an accessible manner, and to help our understanding of the likely impact of results to specific groups of patients. We will communicate directly with the National Institute for Health and Care Excellence to ensure they are kept informed of results that are of direct relevance to the guidance they have issued on COPD, and with the Medicines and Healthcare Products Regulatory Agency if it appears that findings may impact the risk/benefit profile of COPD treatments.

**Author affiliations**
[1]Department of Non-communicable Disease Epidemiology, Faculty of Epidemiology and Population Health, London School of Hygiene and Tropical Medicine, London, UK
[2]Department of Medical Statistics, Faculty of Epidemiology and Population Health, London School of Hygiene and Tropical Medicine, London, UK
[3]Department of Epidemiology, Harvard Medical School, Boston, Massachusetts, USA
[4]Division of Pharmacoepidemiology and Pharmacoeconomics, Department of Medicine, Brigham and Women's Hospital, Boston, Massachusetts, USA
[5]National Heart and Lung Institute, Imperial College London, London, UK

**Contributors**  KW, EW, JRC, LW, SS, LS, JKQ and ID contributed to study question and design. KW wrote the first draft of the protocol manuscript (based on original grant/scientific approval applications to NIHR and ISAC that KW, EW, JRC, LW, SS, LS, JKQ and ID all contributed to). KW, EW, JRC, LW, SS, LS, JKQ and ID contributed to further drafts and approved the final version.

**Funding**  This work was supported by NIHR grant number 15/80/28.

**Competing interests**  JRC is funded 60% by a grant from the MRC, via a secondment to the MRC Clinical Trials Unit. The remaining funding is a combination of HEFCE and small contributions from various NIHR, ESRC, MRC and EU funds. He undertakes methodological consultancy work for Novartis and GlaxoSmithKline (GSK), and has given missing data courses for GSK, Bayer and Boehringer. Professor Smeeth reports grants from Wellcome, MRC, NIHR, BHF, Diabetes UK, ESRC and the EU; grants and personal fees for advisory work from GSK, and personal fees for advisory work from AstraZeneca. He is a Trustee of the British Heart Foundation. LW is an independent consultant to the pharmaceutical industry and is employed to provide advice by a number of different companies, none of which is involved in this therapeutic area. SS is a consultant to WHISCON and to Aetion, a software manufacturer of which he also owns equity. He is principal investigator of investigator-initiated grants to the Brigham and Women's Hospital from Bayer, Genentech and Boehringer Ingelheim unrelated to the topic of this study. He does not receive personal fees from biopharmaceutical companies. LS reports grants from Wellcome, MRC, NIHR, BHF, Diabetes UK, ESRC and the EU; grants and personal fees for advisory work from GSK, and personal fees for advisory work from AstraZeneca. He is a Trustee of the British Heart Foundation. JKQ research group has received funding from MRC, Wellcome, BLF, GSK, BI, AZ and Insmed for other projects, none of which relate to this work. ID is funded by an unrestricted grant from, has consulted for and holds stock in GSK.

**Patient consent**  Detail has been removed from this case description/these case descriptions to ensure anonymity. The editors and reviewers have seen the detailed information available and are satisfied that the information backs up the case the authors are making.

**Ethics approval**  Ethical approval for this study has been obtained from the London School of Hygiene & Tropical Medicine Ethics Committee (Ref: 11997) and the cli nicalstudydatarequest.com review panel.

**Provenance and peer review**  Not commissioned; externally peer reviewed.

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
