## [Reviewer comments · BMJ Open]

ARTICLE DETAILS

TITLE (PROVISIONAL)	Real world effects of medications for chronic obstructive pulmonary disease: protocol for a UK population-based non-interventional cohort study with validation against randomised trial results
AUTHORS	Wing, Kevin; Williamson, Elizabeth; Carpenter, James; Wise, Lesley; Schneeweiss, Sebastian; Smeeth, Liam; Quint, Jennifer; Douglas, Ian

VERSION 1 – REVIEW

REVIEWER	Tetyana Kendzerska The Ottawa Hospital Research Institute, University of Ottawa, Canada
REVIEW RETURNED	01-Oct-2017

GENERAL COMMENTS	Thank you for the opportunity to review this nicely written protocol by Wing and co-authors titled “Real world effects of medications for chronic obstructive pulmonary disease: protocol for a UK population-based observational cohort study with validation against randomised trial results”. COPD is chronic prevalent and incurable condition. As authors mentioned, COPD treatment guidelines are largely informed by randomised controlled trial results, but it is unclear from current evidence if these findings apply to large patient populations not studied in trials. I agree that properly done observational studies could be very useful to study patient groups excluded from trials. However, I have some concerns with the protocol as written: Abstract: (1) The dates of the study should be included in the “Methods and analysis” subsection(2) Details on validated methods for detecting COPD within the Clinical Practice Research Datatlink would be useful in abstract. Introduction: (1) Please provide references for the sentence: “Observational data will be obtained from very large UK population-level databases of electronic health records.” Method section: (1) Although authors mentioned that “Validated algorithms have been established to identify COPD, COPD exacerbations and pneumonia (both hospital and primary care managed) in CPRD/HES linked data”, they did not provide the exact definitions, just references. Given that COPD is the condition of interest, and COPD exacerbations and pneumonia are outcomes, details on definitions
---

	would be helpful to understand potential source of misclassification bias, even as a separate table or in the data supplement. Please also provide sensitivity and specificity of definitions. (2) The dates of the study, cohort creation using health administrative data, should be included. (3) Authors mentioned that “Feasibility work shows there are ~11,000 TORCH eligible patients in CPRD who did not receive FP/SAL at the time they attained TORCH eligibility and therefore have at least one-time period that means they are eligible for inclusion as an Objective 1 unexposed participant” – not clear within which time frame. (4) In Step 3 authors presented characteristics for matching – among comorbidities only history of cardiovascular disease seems will be considered. Would it be important to consider the Charlson Comorbidity Index or the Johns Hopkins Aggregated Diagnosis Groups? (5) If authors want to evaluate the treatment effects in patients excluded from trials, why individuals with a history of asthma would be excluded? (6) A reference need to be provided for the following sentence “We have substantial prior experience of building propensity models, and our study team includes the lead of an MRC funded project to determine optimal propensity score methods for use with missing data (EW).” (7) Details on why 1:1 matching was chosen and the definition of “the closest propensity score” would be beneficial. (8) How comparable are definitions of COPD and outcomes between the TORCH and the proposed cohort study? (9) Would it be important to mention misclassification bias given that the proposed study will be based on health administrative data?
--	---

REVIEWER	Anders Løkke Department of Respiratory Diseases and Allergy, Aarhus University Hospital, Noerrebrogade 44 8000 Aarhus C, Denmark
REVIEW RETURNED	02-Nov-2017

GENERAL COMMENTS	My overall comment is, that the manuscript needs a major revision before being able to evaluate it in details and giving it a yes or no (I think it will be a yes, though). The manuscript is written in plain English. However, the structure of it, makes it rather difficult to follow - and it is extremely hard to keep track because the authors keep on mixing together the two/three objectives, that they want to look at in the study. The general idea of the study is good and I find it relevant and doable. The structure of the manuscript needs to be much more precise and strict in order to make it understandable to others. Also - the two/three steps described in the study need to be separated completely - and to be described one by one in order to give an understandable overview for the reader. A nice figure showing the two/three steps in details - again - separated completely into step one, step two and step three is also much appreciated Moreover I am not sure that I fully understand what and why you are doing?!
---

Objective1:

For instance - why is it of interest to simulate TORCH in real life (I for one am not sure) ?

Also it is very difficult to get from the protocol the timeperiod for these patients - what is the minimum allowed time period on study medication (6 month, a year, 3 years like Torch)? - If not 3 years you are not replicating the study but trying to mimic by statistical support and this is of no use....

Another problem when trying to understand the results is, that people do not seem to drop out of your study (because of the way they are selected) - they did in TORCH.

Compliance is addressed in the end of the protocol - however, in my point of view this is extremely relevant - if compliance differs from TORCH - and it most likely will - you are not replicating the study and you cannot adjust the results properly.

Also, I am not sure of the "no medication group" how are they generated - from the database as well, how are they defined - no medication ever or only in a certain time window (6 months, a year etc.) This could potentially be very biased as they could receive medication from another place or have a lot of medication in stock?!

I cannot read from the manuscript the time-period you are exploring - a lot has happened since TORCH - focus on smoking (legislation, taxes, bans etc.), rehabilitation, more focus on COPD and inhalation medication, vaccinations etc. Therefore, it is very difficult to compare data in an unbiased way.

Finally I am not keen on the first step because you are on a retrospective fishing-trip with statistical support - if you want to know TORCH results in a real-world setting -- you should do TORCH over again - in a Real-world setting. If not - do not go there - that is my advice.

Objective 2

This is more like it. This is interesting!

However, you should describe more in details how and why.

Again, the time period on medication is very interesting (6, 12, 24 months etc.) and the time period of the data-span. Compliance and drop out of medication as well!

It can be difficult to interpret in full - especially if the extracted data is made up of different data (6 months on medication from patient A, 2 years made up from two different 1 year time-periods for patient B and three years in a row for patient c) - then it can be very difficult to call it 2 years follow-up data as this is clearly not the case.

Objective 3:

I wonder why patients are allowed to have 1 exacerbation in this mild group? Even in exacerbation-rich studies they have only 1 - or even less (Flame, TOCH etc.). Therefore, one could argue that these patients are not mild when exacerbating. I suggest that this is removed.

In summary:

This protocol is of interest - for sure!

However, it faces some serious challenges.

I will advise you to skip objective 1 as this is a pseudo-comparison of

	very little interest (and it cannot be done this way). Objective 2 and 3 are much more doable - however you need to describe in detail (as mentioned) how and why - complementary figures and timelines are much appreciated. I strongly recommend a revision to begin with. Best of luck - looking forward to the revision.
--	--

VERSION 1 – AUTHOR RESPONSE

#	Feedback	Response and details of any changes made
Author-driven changes to this version		
1.	We have decided to change the term “observational” where it is used throughout the study to “non-interventional”. So (e.g.) “observational study” becomes “non-interventional” study. We consider this to be a better way of distinguishing the type of data we are using (and study we are performing) from RCT data, because within an RCT, after randomisation to the intervention participants are also “observed”.	21 changes throughout the manuscript, changing the term “observational” to “non-interventional” (including the title). We have also added the following text to the first sentence of the third paragraph of the Background and rationale section: “(sometimes also referred to as “observational studies”)
2.	We have decided to add a number of additional exclusion criteria that are included in the TORCH study protocol but were not specified in the published TORCH article. These relate to previous long-term oxygen therapy, presence of diagnoses likely to interfere with the study or cause death within 3 years, current use of oral corticosteroid therapy, and exacerbations during a 2-week run-in period. When applied to the cohort, these additional criteria have only a small impact on the final numbers, so the approximate feasibility counts provided throughout the article remain unchanged. We have also decided to reorganise the list of inclusion and exclusion criteria for clarity, in particular delineating which criteria are inclusion and which are exclusion.	Changes to the Selection of participants section, Objective 1, Step 1 so that it now reads as follows: “We will select all patients in the CPRD with COPD who are eligible for HES-linkage and during the period covered by the linkage would have met the following TORCH study inclusion criteria:  • a diagnosis of COPD • age 40-80 years • smoking history • lung function (FEV₁ <60% predicted, FEV₁/FVC ratio <70%) An eligible-for-inclusion date will then be assigned as the date that all of the above inclusion criteria were met for the individual. We will then exclude any individual who has any of the following TORCH study exclusion criteria prior to their eligible-for-inclusion date:  • a diagnosis of asthma (within the previous 5 years)

#	Feedback	Response and details of any changes made
		 • a diagnosis for any (non-COPD) respiratory disorder • a record of lung surgery • a diagnosis of alpha-1 antitrypsin deficiency • evidence of drug or alcohol abuse • a record of having received long-term oxygen therapy • diagnoses for conditions likely to interfere with the TORCH trial or cause death within 3 years • current use of oral corticosteroid therapy (defined as continuous use for greater than 6 weeks, with courses of oral corticosteroid separated by a period of less than 7 days considered as continuous use) • any exposure to FP/SAL within the previous 4 weeks Finally, in-line with the TORCH trial approach, anyone who has an exacerbation requiring oral corticosteroid therapy or hospitalisation during the run-in period (the 2-week period following eligibility) will also be excluded.”
Reviewer 1 (Tetyana Kendzerksa)		
Abstract		
1	The dates of the study should be included in the “Methods and analysis” subsection	We have updated the Methods and analysis section of the Abstract with the text “(selecting people from between the dates of 1 st January 2004 and 1 st January 2017)”.
2	Details on validated methods for detecting COPD within the Clinical Practice Research Datalink would be useful in abstract.	We have updated the Methods and analysis section of the Abstract with the text: “and validated methods for detecting COPD and COPD exacerbations in routinely collected primary care data”.
Introduction		
1	Please provide references for the sentence: “Observational data will be obtained from very large UK population-level databases of electronic health records.”	We have replaced the second half of this sentence with “the UK Clinical Practice Research Datalink (linked to the Hospital Episodes Statistics database)” and added a reference.
Method		
1	Although authors mentioned that “Validated algorithms have been established to identify COPD, COPD exacerbations and pneumonia (both hospital and primary care managed) in CPRD/HES linked data”, they did not provide the exact definitions, just references. Given that COPD is the	While reviewing the manuscript in response to this point, we noticed an updating error in relation to which algorithms have been validated, and have therefore updated the quoted sentence (at the end of the Setting/Data Sources – CPRD section) to read:

#	Feedback	Response and details of any changes made
	condition of interest, and COPD exacerbations and pneumonia are outcomes, details on definitions would be helpful to understand potential source of misclassification bias, even as a separate table or in the data supplement. Please also provide sensitivity and specificity of definitions.	“Algorithms have been established to identify COPD, COPD exacerbations and pneumonia (both hospital and primary care managed) in CPRD/HES linked data (including validated algorithms for COPD and exacerbations).^{10-12,} We have also added a table with a high level overview of each algorithm to the supplementary materials (including reported measures of validity), and included the following additional text after the sentence above in the Setting/Data Sources – CPRD section: “See supplementary materials for a high-level overview of these algorithms.”
2	The dates of the study, cohort creation using health administrative data, should be included.	We have added the text “from between the dates of 1st January 2004 and 1st January 2017” to the first paragraph of the Selection of participants section.
3	Authors mentioned that “Feasibility work shows there are ~11,000 TORCH eligible patients in CPRD who did not receive FP/SAL at the time they attained TORCH eligibility and therefore have at least one-time period that means they are eligible for inclusion as an Objective 1 unexposed participant” – not clear within which time frame.	We have amended this sentence in the Selection of participants, Objective 1, Step 2 section so that it reads: “Feasibility work shows that between 1st January 2004 and 1st January 2017 there were ~11,000 TORCH eligible patients in CPRD who did not receive FP/SAL at the time they attained TORCH eligibility and therefore have at least one time period that means they are eligible for inclusion as an Objective 1 unexposed participant.”
4	In Step 3 authors presented characteristics for matching – among comorbidities only history of cardiovascular disease seems will be considered. Would it be important to consider the Charlson Comorbidity Index or the Johns Hopkins Aggregated Diagnosis Groups?	Although the Charlson Comorbidity Index or the John Hopkins Aggregated Diagnosis Groups would be a better way of matching people with COPD from CPRD to those included in the TORCH trial than cardiovascular disease alone, we are limited as to which comorbidities were recorded as part of the TORCH study. For example, although the renal and liver disease outcomes required as part of the Charlson Comorbidity Index could be obtained from CPRD, a history of renal or liver conditions was not collected by TORCH.

#	Feedback	Response and details of any changes made
		A history of cardiovascular disease was captured by TORCH, however, and we therefore chose this as a marker of comorbidity on which to match to.
5	If authors want to evaluate the treatment effects in patients excluded from trials, why individuals with a history of asthma would be excluded?	In Objective 1, our aim is to create a TORCH-analogous cohort that we can then analyse to see if we are able to replicate the TORCH results. We exclude those with asthma from this cohort inline with the TORCH exclusion criteria. For our objective 3 analysis where we aim to study people with milder COPD, our aim is still to remove those with evidence of concurrent COPD and asthma, in order to ensure that we are studying a homogenous population of those with milder COPD. We agree with Reviewer 1 that it would be of interest to study people with asthma and COPD, however, as these individuals would have been excluded from TORCH. We think that it would be most appropriate to include this aim as part of Objective 2, so have added the following item to the list of additional criteria (in the first sentence of Selection of participants, Objective 2 heading): “(4) an asthma diagnosis at any time prior to inclusion”. In reviewing the manuscript and TORCH protocol in relation to this point, we have also made the following updates to the manuscript.  1. In the TORCH protocol, the asthma-related exclusion specifies those with “a current diagnosis of asthma”. Our current equivalent Objective 1 criteria is to exclude anyone with a “history of asthma”, which we think is likely to be more restrictive than the TORCH approach. We have therefore modified this (in Selection of participants, Objective 1, Step 1) so that it reads: “a diagnosis of asthma (within the previous 5 years)” and added a sentence: “Given the limited information on how asthma exclusions were applied in the TORCH study, we will perform a sensitivity analysis in which the asthma exclusion is a diagnosis within the previous 1 year, rather

#	Feedback	Response and details of any changes made
		than 5 years as specified above.” 2. For our Objective 3, we specify “no evidence of asthma” which we feel is confusing, because we already specify that the listed Objective 3 criteria are “additional” (to Objective 1), and Objective 1 already describes how those with asthma will be excluded (see point 1. above). We have therefore removed the text “AND no evidence of asthma” from the Selection of participants, Objective 3 section.
6	A reference need to be provided for the following sentence “We have substantial prior experience of building propensity models, and our study team includes the lead of an MRC funded project to determine optimal propensity score methods for use with missing data (EW).”	Four reference citations have been added to support the points made in this sentence in the Statistics analysis section (Propensity score for addressing confounding subheading, last sentence of paragraph 1). One of the citations is to a reference already included in the reference list (ref 16), while the other three are to newly added references (refs 13-15).
7	Details on why 1:1 matching was chosen and the definition of “the closest propensity score” would be beneficial.	At the end of the Methods of analysis section we state that we will compare the characteristics of the propensity score matched FP/SAL treated and untreated group, and then go onto state that “if there are substantial differences noted for important variables, it may be necessary to further adjust the statistical models.¹⁷”, referencing the paper by Austin on the use of Propensity Score Methods for reducing the effects of confounding in observational studies. We acknowledge that this step could also include examining the way in which we attempt to obtain the closest propensity score match (i.e. type of matching), and that the matching ratio we use may also influence our ability to match (points also covered in the Austin paper). In response to this point from Reviewer 1 and to clarify this, we have added the following additional sentence at the end of this section: “This could also include examining the effect of using greedy vs optimum matching approaches in order to obtain the closest propensity score match, and/or matching at a ratio other than 1:1.”
8	How comparable are definitions of COPD and outcomes between the TORCH and the proposed cohort study?	It is difficult to be completely certain that the definitions are comparable, given what can be scarce information available in the TORCH trial.

#	Feedback	Response and details of any changes made
		Our algorithm for selecting people with COPD from CPRD has been validated against diagnoses made by GPs in UK primary care (across a period of time that included when the TORCH trial was performed), however, and we do not think that a COPD diagnosis would differ substantially for a trial participant compared to a GP patient. Furthermore, the inclusion criteria for TORCH includes (objective) age, smoking and lung function criteria which we also apply to our own cohort (as detailed in the Selection of participants section, Objective 1, Step 1), which increases our confidence that the participants will actually be people with COPD. Our outcome definitions for COPD exacerbation and pneumonia use hospital episode statistics as well as GP data, and again have been validated against diagnoses made by GPs in UK primary care. We also do not think that these outcomes would differ substantially between a trial and primary/secondary care setting. Any true underlying potential difference is one of the reasons why it might not be possible to replicate trial results for COPD within UK primary care electronic health data, however. If we are unable to obtain similar results to the TORCH trial for our Objective 1 analysis, we will consider the potential role of differences in outcome classification between each setting in any discussion. Please also see the response to point 9 below.
9	Would it be important to mention misclassification bias given that the proposed study will be based on health administrative data?	In response to this point (and in response to point 7 from Reviewer 2) we have included the following new section prior to the Missing data section in the Statistical Analysis section: “Misclassification of (1) drug exposure periods and (2) outcome status It is possible that an individual may still be exposed to FP/SAL for some time after a prescription has

#	Feedback	Response and details of any changes made
		finished, for example if they have medication at home that they haven't used from a previous prescription. This would mean that people may become eligible for inclusion in the unexposed group while they are actually still exposed. If our result differs from the TORCH results (e.g. a rate ratio <0.65 or >0.9), we will conduct a sensitivity analysis in which we include an additional (grace) exposed period equivalent to the length of a single prescription at the end of each actual exposed period, and only classify individuals as eligible for inclusion as unexposed at the end of this additional period. Our results could also be impacted by misclassification of outcome, given the routine nature of the data. Our initial approach for detection of COPD exacerbations is to use a validated case definition from previous work that maximises positive predictive value while maintaining a relatively high sensitivity.¹² If our result differs from the TORCH results, we will consider performing a sensitivity analysis in which we assess the impact of applying alternative case definitions for COPD exacerbations (see supplementary materials for an overview of articles relating to the case definitions we plan to utilise, including any validity measurements provided).”
Reviewer 2 (Anders Løkke)		
General		
1	The manuscript is written in plain English. However, the structure of it, makes it rather difficult to follow - and it is extremely hard to keep track because the authors keep on mixing together the two/three objectives, that they want to look at in the study. The structure of the manuscript needs to be much more precise and strict in order to make it understandable to others. Also - the two/three steps described in the study need to be separated completely - and to be described one by one in order to give an understandable overview for the reader.	We appreciate this point from Reviewer 2 in relation to the overall structure of the protocol. The main differences between the three objectives relate to the Selection of participants section, which is already organised by objective. There are only three other (relatively short) sections that differ by objective: Exposures, Sample size, and Propensity score for addressing confounding. The remaining fourteen headings in the protocol are applicable to all objectives. A complete separation of the objectives within the article would therefore mean that the majority of sections would be repeated, which we feel would detract from the clarity of the manuscript, and significantly increase its length. We do agree that the organisation of three of the four sections that differ by objective could be improved, and in response to this point have amended these as detailed below.

#	Feedback	Response and details of any changes made
		Selection of participants Figures 2 and 3 (which were figure 1 and 2 in the original, see response to point 2 below) relate to the selection of participants for objective 1, but were placed after the objective 3 subheading. We have moved these to be placed immediately after the objective 1 subheading for clarity. Exposures In the original paper the 2nd and 4th paragraphs related to objective 1, with the 3rd and 5th relating to objectives 2 and 3. We have now rearranged this section so that all content related to objective 1 is discussed before any of the content pertaining to the other objectives. We have also replaced the sentence “For Objectives 2 and 3, exposures are as follows:” with “Specific exposures for Objectives 2 and 3 are as follows:”. Sample size We have added subheadings per objective, and removed the text “for each objective” from the second sentence. We have also added the text “as part of Objective 1” to the opening sentence of the Validation of results against TORCH section (Statistical analysis) in order to clarify it relates to objective 1.
2	A nice figure showing the two/three steps in details - again - separated completely into step one, step two and step three is also much appreciated	We have included a new Figure 1 within the Methods and analysis section (immediately after the Setting/data sources section) that clearly describes the separate steps (objectives) and how they relate to each other, as well as illustrating the data sources used (“Figure 1: Overview of study objectives and sources of data for the COPD real-world medicines effects study”). We have also added the following paragraph immediately after the Methods and analysis heading: “Figure 1 provides a high-level overview of the study, detailing each objective and data source used, and showing how existing RCT data will be used in Objective 1 in order to validate methods for analysing COPD in routinely collected electronic

#	Feedback	Response and details of any changes made
		health data that will then be applied to unanswered questions in Objective 2 and 3.”
3	Moreover I am not sure that I fully understand what and why you are doing?!	We hope that the new Figure 1 referred to above helps communicate what we plan to do in this study, and why we are doing it, particularly if considered with the information already provided in the Background and rationale section (specifically paragraphs 2 – 4). Please also see response to point 4. below.
Objective 1		
4	For instance - why is it of interest to simulate TORCH in real life (I for one am not sure) ?	We would refer Reviewer 2 to paragraph 3 of the Background and rationale section, which explain that:  (1) while the use of non-interventional studies for studying drug harms is well established, their use for estimating treatment effectiveness is not. (2) it is of key importance to be able to demonstrate that methods for analysing treatment effects using non-interventional data are able to generate results that can be trusted. In response to this point we have added the following text to the end of paragraph 3 of the Background and rationale section, so that it is clear exactly how the aims and objectives of this study relate to the points already made in this paragraph. “For example, the availability of anonymised individual patient data from randomised controlled trials provides the potential for “RCT-analogous” cohorts to be selected from non-interventional data sources (by matching patient records from non-interventional data to the RCT patient records on key characteristics). If subsequent analysis of a non-interventional RCT-analogous cohort generates results that are similar to those generated by the reference RCT, one could be confident in the validity of the results, and in the non-interventional methods used to obtain the results.”
5	Another problem when trying to understand the results is, that people do not seem to drop out of your study (because of the way they are selected) - they did in TORCH.	We acknowledge that there are likely to be a number of differences between the TORCH-analogous cohort that we select as part of Objective 1 and the actual TORCH trial population (see paragraph 3 of the Objective 1 – Step 3 subheading

#	Feedback	Response and details of any changes made
		of the Selection of participants section). It is not our aim to create a cohort that is exactly the same as TORCH, and we do not believe that this would be possible, given the routine nature of the data in CPRD and the fact that there may be determinants of patient selection and reasons for drop-out from the TORCH trial that we may never know. For example, one of the reasons for drop-out specified in the TORCH trial is “Other reasons”. There will also be reasons why people in CPRD drop out of our analysis that would not relate to TORCH (for example, they move GP practice or drop out of the database for another reason). In response to this point, we have updated the CPRD paragraph in the Setting/data sources with the following additional sentence, so that it is clear that people may also drop out of cohorts selected from CPRD: “A patient starts contributing follow-up time to the database at the date they join an “up to standard” practice (or the date that their practice starts contributing up to standard data), and stop contributing follow-up time on either their death date, their transfer out date (the date that they leave the database due to reasons other than death) or on the last collection date for their practice.”
6	Compliance is addressed in the end of the protocol - however, in my point of view this is extremely relevant - if compliance differs from TORCH - and it most likely will - you are not replicating the study and you cannot adjust the results properly.	We have included the Handling measurement of adherence to medication in the Statistical analysis section as we believe this is the most appropriate location for it. In terms of where the Statistical analysis section has been placed in the protocol – we have followed STROBE guidelines for which sections to include and their ordering. We certainly agree with Reviewer 1 that compliance is extremely relevant to the study, which is why the Handling measurement of adherence to medication section is the largest single section under the Statistical analysis heading, and also why assessment of adherence is listed as a limitation in the Strengths and limitations section.

#	Feedback	Response and details of any changes made
		Furthermore, within the Handling measurement of adherence to medication we state that “the data analysis for adherence will necessarily be a significant element of the work to be done for this study”. We also detail pilot work already performed to assess FP/SAL adherence within CPRD that shows that overall adherence is poorer in CPRD than it was for TORCH, but that there is a wide range of adherence in routine care which will allow us to perform a (pre-defined) sensitivity analysis if our results for Objective 1 show a null or poorer treatment effect than TORCH, restricting to those people with similar adherence to TORCH. We are therefore of the opinion that we have already provided serious and adequate consideration to the issue of compliance/adherence, and have not made any additional changes to the protocol in response to this point.
7	Also, I am not sure of the "no medication group" how are they generated - from the database as well, how are they defined - no medication ever or only in a certain time window (6 months, a year etc.) This could potentially be very biased as they could receive medication from another place or have a lot of medication in stock?!	The definition of our unexposed group is provided in the Selection of participants section, Objective 1. Under the Step 1 heading, it is stated that participants will be selected from CPRD based upon the TORCH inclusion and exclusion criteria, which includes a requirement that they should not have had “any exposure to FP/SAL within the previous 4 weeks”. Step 2 then describes that “during any time between attaining TORCH eligibility and a subsequent prescription for FP/SAL, patients will be eligible as an unexposed patient in Objective 1.” and also that “there may be multiple time periods within a person’s record where eligibility as an unexposed patient is met”. These statements are further supported by the example eligibility timeline for a person in figure 3, which specifically illustrates how the unexposed periods are defined, and contains extensive footnotes that further elaborate on eligibility and selection of unexposed participant-time. We consider this to be a detailed and clear description of how unexposed people are selected from CPRD in our study, and have therefore not made any changes to the text in response to this comment. While reviewing the manuscript in relation to this point, we did, however, update the title of figure 3 to make the figure more standalone by

#	Feedback	Response and details of any changes made
		clarifying that “unexposed/exposed” equates to “FP/SAL untreated/treated”. We also made a very minor amendment to figure 3 (moving the “1 week” key so that it labelled one of the weeks in the diagram itself). We agree that accounting for this possibility would improve the study. In response to this point and the (Method) point 9 from Reviewer 1, we have created a new section (see Reviewer 1: Method – point 9 response for details).
8	I cannot read from the manuscript the time-period you are exploring – a lot has happened since TORCH - focus on smoking (legislation, taxes, bans etc.), rehabilitation, more focus on COPD and inhalation medication, vaccinations etc. Therefore, it is very difficult to compare date in an unbiased way.	In response to this point (and Reviewer 1: Method – point 2) we have included the following additional text in the first line of the Selection of participants section: “from between the dates of 1st January 2004 and 1st January 2017”. The dates provided above show that the period we are considering includes the TORCH study publication date (2007), in addition to three years prior to this date and a substantial period following TORCH. We will therefore be selecting people from a range of COPD clinical care scenarios (including COPD care prior to 2007, and COPD care following 2007). Restricting selection of people so that eligibility had to be within the same time period as TORCH would reduce our sample size, and we believe that the matching we will perform by age, sex, body mass index, previous COPD medication treatment, COPD exacerbation history, history of cardiovascular disease and lung function is likely to generate a cohort with similar characteristics to the TORCH participants (and that these factors will be more important than date of recruitment). We have not made any changes to the manuscript in response to this point.
9	Finally I am not to keen on the first step because you are on a retrospective fishing-trip with statistical support - if you want to know TORCH results in a real-world setting -- you should do TORCH over again - in a Real-world setting. If not - do not go there - that is my advice.	Objective 1 does not describe a “fishing trip with statistical support” but a pre-defined approach for creation of a TORCH-analogous dataset, and a clear pre-defined approach for analysis of this dataset. This includes details of the statistical techniques to be applied, a clear definition of what would be considered a “similar” result to the TORCH

#	Feedback	Response and details of any changes made
		trial, and carefully considered planned sensitivity analyses. The aim of this study is to provide evidence as to whether it is possible to use non-interventional data and methods for studying treatment effects of COPD, and if it is, to use the validated methods to study unanswered questions. It is not to repeat TORCH, but to:  (1) improve understanding as to the utility of non-interventional data and methods for studying COPD treatment effects using TORCH as a validation tool before (2) using these validated methods to study unanswered questions. Without the successful completion of Objective 1, we would not have any confidence in findings for Objectives 2 and 3, and therefore Objective 1 is an essential component of the study. We think that the manuscript accurately conveys these aims, particularly with the changes made in the responses to points 1 to 8 from Reviewer 2, and have not made any further changes in relation to point 9.
Objective 2		
10	This is more like it. This is interesting! However, you should describe more in details how and why.	We are happy that Reviewer 2 appreciates our Objective 2 and finds it interesting. However, the central concept of our study is that it is not known if methods for studying COPD treatment effects in non-interventional data are valid. Therefore in our Objective 1 we aim to validate these methods by creating a study population that is analogous to an RCT population and seeing if we obtain similar results to the RCT if we analyse this RCT-analogous population. Once we have done this, then we can study unanswered questions (like those detailed in Objective 2) with the knowledge that the results we obtain are likely to be valid. We hope that the changes made in the answers above have clarified this for Reviewer 2, and also serve to highlight why Objective 2 would be of far less interest if we hadn't first validated our methods as part of Objective 1. In relation to "why" we are including an Objective 2

#	Feedback	Response and details of any changes made
		to measure the effect of COPD treatment effects in patients excluded from trials: this is because COPD treatment guidelines are largely informed by randomised controlled trial (RCT) results, but it is not clear if these findings apply to large patient populations not studied in trials. FP/SAL is one of the most widely used COPD treatments. It was studied in large randomised trials (e.g. the TORCH trial), but the effects of treatment in important patient groups who were not studied are unknown, meaning that conclusions about these groups are difficult to make. Additional studies within these groups are therefore needed. We believe this is a clear and detailed justification of why we are carrying out Objective 2, and as it is already clearly laid out in this way in the Background and rationale section, we have not made any changes in relation to this point. In relation to the “how” we are doing this: as mentioned in the response to point 1 above, the majority of the manuscript consists of headings that apply to all Objectives. Therefore details of how we are implementing Objective 2 are provided in the following sections: Study design, Setting/data sources (CPRD subheading), Selection of participants (Objective 2 subheading), Exposures, outcomes and co-variates, Outcomes, Samples size, and the Statistical Analysis sections. Upon reviewing the manuscript in relation to this point, we decided to include the following additional detail about the selection of participants for both Objective 2 and Objective 3, which we hope will help further clarify the “how” for both objectives: Selection of participants Objective 2: Measurement of COPD treatment effects in patients excluded from trials (text added to end of section) “Participants for Objective 2 will be selected in a similar fashion to Objective 1, with the additional eligibility criteria mentioned above applied (i.e. a modified Step 1). As for Objective 1, each participant will be allowed to have multiple FP/SAL

#	Feedback	Response and details of any changes made
		exposed and unexposed eligibility periods in their record, as described in Figure 3. In contrast to Objective 1, there will be no matching of unexposed patients to TORCH patients, as we do not require a TORCH-analogous cohort for this analysis (i.e. no Step 3). All other selection steps will be as applied for Objective 1, including the use of propensity score matching in order to obtain comparable unexposed and exposed groups for analysis.” Objective 3: Determination of treatment effects in an under-studied disease stage (text added to end of section) “As for Objective 2, the selection steps will be similar to Objective 1, with modified criteria for step 1 and the removal of the TORCH-matching step (step 3).”
11	Again, the time period on medication is very interesting (6, 12, 24 months etc.) and the time period of the data-span. Compliance and drop out of medication as well!	(a) Time period on medication Please see the response to point 7 above, that highlights the sections of the text that relate to definition of unexposed and exposed periods in Objective 1, and also the response to point 10 above that clarifies that the approach for selection of unexposed/exposed participants for Objective 2 is the same as Objective 1. (b) Time period of the data span Please see the response to point 8 above. (c) Compliance and drop out of medication Please see the responses to point 6 and 7 (second half) above.
12	It can be difficult to interpret in full - especially if the extracted data is made up of different data (6 months on medication from patient A, 2 years made up from two different 1 year time-periods for patient B and three years in a row for patient c) - then it can be very	Our response to this point assumes that Reviewer 2 means 3 years follow-up (which is what both TORCH and our study specify) and not 2 years follow-up. All TORCH trial analyses were performed according

#	Feedback	Response and details of any changes made
	difficult to call it 2 years follow-up data as this is clearly not the case.	to the intention-to-treat principle and the analyses that we are performing will also follow this principle. This means that once the follow-up for either an unexposed or exposed participant has begun in our study, we will follow them up from their unexposed or exposed start date for a period of 3 years (or until their death or other reason for exiting the database) and they will remain assigned to the same exposure category for their entire follow-up period, irrespective of whether they actually change status. Therefore, describing it as 3 years follow-up is accurate. We have updated the Statistical analysis section, Methods of analysis subheading with the following text in response to this point (inserted after the first sentence): “All analyses will be performed according to the intention-to-treat principle (as was done in the TORCH study), meaning that if a participant enters the study as either an exposed or unexposed participant, they will remain assigned to that exposure category for the entire duration of their follow-up (irrespective as to whether their true exposure status changes).”
Objective 3		
13	I wonder why patients are allowed to have 1 exacerbation in this mild group? Even in exacerbation-rich studies they have only 1 - or even less (Flame, TOCH etc.). Therefore, one could argue that these patients are not mild when exacerbating. I suggest that this is removed.	We appreciate this point from Reviewer 2. We would like to include the possibility of analysing people who have better lung function than those recruited to TORCH, but might end up having an exacerbation within 1 year. If these people are not included, the effect of the medication on this group remains unknown. In response to this point, we have updated the end of the first sentence of Selection of participants, Objective 3 to read: “(2) no exacerbations in the year post COPD diagnosis.” and added an additional sentence immediately following this that reads: “We will also perform a sensitivity analysis where we allow the group of people with FEV1 >60% predicted who had a maximum of one exacerbation within 1 year post COPD diagnosis to be included.”

#	Feedback	Response and details of any changes made
Generally applicable		
15	This protocol is of interest - for sure! However, it faces some serious challenges. I will advise you to skip objective 1 as this is a pseudo-comparison of very little interest (and it cannot be done this way). Objective 2 and 3 are much more doable - however you need to describe in detail (as mentioned) how and why - complementary figures and timelines are much appreciated.	We hope that the points made above and changes to the manuscript have conveyed why Objective 1 is an essential part of this study, and provided some additional clarity around the overall project and the details of Objectives 2 and 3.

VERSION 2 – REVIEW

REVIEWER	Anders Løkke Department of Respiratory Diseases and Allergy, Aarhus University Hospital, Noerrebrogade 44, 8000 Aarhus C
REVIEW RETURNED	20-Dec-2017

GENERAL COMMENTS	Thank you for letting me review this manuscript once again. It has been much improved and it far better now. However, I do have some points for clarification/further improvement before accepting it: It seems as the patients selected are the same in all 3 objectives with some minor variations – is this correct? If so this should be specified (for details look below). Objective 1 In this study, you only have 2 groups (treated and untreated) – is this correct – and treated means Seretide and untreated means no Seretide?! What about (especially) LAMA (more and more COPD patients are treated with LAMA as time advances), other ICS/LABA (Symbicort and others) etc. – can they have this one the side or not and what about the untreated can they have every other medication or not?! This have to be described in further details – I for one simply cannot read it out of your paper?
---

In TORCH, approx. 60% got medication prior to run in, that is 22% ICS, 8% LABA and 30% ICS/LABA – 40% received no medication. This is nice information when looking at the results. Ideally, they should be matched on this (but it might be close to impossible) – however, if they all receive ICS/LABA it can be because they are sicker than in TORCH (and not just randomized by chance) which might influence the result (the doctor chose the medication for a reason)?

How will you evaluate the effects, are they all equally important and how close to TORCH do they have to be to be called the same – could you elaborate?

I think it would be wise and fair to track how many people that change medication during the study (it is a lot I can assure you) – for instance compare beginning of the 3 years with the end – especially if your results differ from TORCH this might be the most obvious reason (as well as the once described above).

Objective 2

As I read it – again you only look at Seretide in the exposed group and no Seretide in the unexposed group?! – this time in a larger group (also including oxygen, comorbidities etc.) – or what (or is it only in oxygen, comorbidities etc.)? You write additional criteria but is it additional to the diagnosis (I think so) or to the criteria mentioned in objective 1 – please specify as you write about objective 1 in objective 2, so this is unclear at the moment.

As I see it you have the same medication issues here as in objective 1.

I suppose you will look at the same outcome parameters?

Objective 3: See objective 2. Identical questions.

Exposures

You write about secondary analysis. I am not sure what you mean/how you will do this with respect to medication – this is baseline information I suppose? Why is no medication only present under objective 2 and 3 (the untreated in objective 1 are not truly untreated or?)

	Finally, my biggest concern: You write under handling measurement of adherence to medication, that you have performed a sample on 30 patients over 1 year and found 2/3 were still receiving Seretide. Try to repeat that for 2 and 3 years – you will indeed be surprised to see how dramatically that number drops. I have done a lot of database research with medication and it is NOT easy! 1 year studies are ok but the longer the period – the more disappointing and confusing the results. I wish you the best of luck.
--	---

REVIEWER	Tetyana Kendzerska University of Ottawa, Ottawa Hospital Research Institute, Ottawa, Ontario, Canada
REVIEW RETURNED	05-Jan-2018

GENERAL COMMENTS	The manuscript was revised and improved considerably. All my comments were addressed comprehensively. Thank you again for the opportunity to review this protocol, Tetyana Kendzerska
---

VERSION 2 – AUTHOR RESPONSE

#	Feedback	Response and details of any changes made
Reviewer 2 (Anders Løkke)		
General		
1	It seems as the patients selected are the same in all 3 objectives with some minor variations – is this correct? If so this should be specified (for details look below).	We hope that the answers below have provided clarification.
Objective 1		
2	In this study, you only have 2 groups (treated and untreated) – is this correct –	This is correct – “treated”(exposed) means treated with FP/SAL (Seretide) and “untreated”(unexposed) means not treated with FP/SAL. In response to this

#	Feedback	Response and details of any changes made
	and treated means Seretide and untreated means no Seretide?!	point, we have added the following clarification to the terms exposed and unexposed the first time that they are mentioned in Objective 1. Step 2 paragraph, 2nd sentence now includes the text: “unexposed (to FP/SAL)” Step 4 paragraph, 2nd sentence now includes the text: “exposed (to FP/SAL)” Exposures, outcomes and co-variates section, Exposures heading, first sentence of second paragraph, added: “with FP/SAL” Validation of results against TORCH section, the first sentence now reads: “We will validate our findings against TORCH as part of Objective 1 by determining whether results of the CPRD FP/SAL versus no FP/SAL treatment analysis are compatible with the TORCH exacerbations rate ratio for FP/SAL versus placebo (0.75; 95% CI 0.69-0.81).”
3	What about (especially) LAMA (more and more COPD patients are treated with LAMA as time advances), other ICS/LABA (Symbicort and others) etc. – can they have this one the side or not and what about the untreated can they have every other	TORCH was a placebo-controlled trial, and clearly this is a key difference between our Objective 1 replicated population and analysis and the TORCH trial – we won’t be able to have a group that is prescribed a placebo as a comparison to our FP/SAL treated group.

#	Feedback	Response and details of any changes made
	medication or not?! This have to be described in further details – I for one simply cannot read it out of your paper?	Although we are applying TORCH eligibility criteria that will ensure that patients included in both our unexposed (to FP/SAL) or exposed (to FP/SAL) groups will not be current users of oral corticosteroid therapy (defined as continuous use greater than 6 weeks, see Selection of Participants, Objective 1, Step 1), we will not then be going on to exclude people from either our exposed or unexposed groups based upon COPD medication information from their prescription record entered after their date of recruitment to our study, as this would be using information from “future time” to select participants, which would have not been possible in the TORCH trial and risks biasing our results. We also consider that if we did find a group from primary care records who were eligible for our study but were not on any COPD medications at all, they may well be an unusual group of people, which may also mean we end up with unusual results. We appreciate that this could mean that our groups may differ from the TORCH groups, and in particular our unexposed group may end up containing a greater number of people receiving other COPD medications than the TORCH placebo group did. Note however that in TORCH, both groups were allowed to receive COPD medications for COPD (other than corticosteroids or inhaled long-acting bronchodilators) as required and in this respect our design has some similarity. If we find that we are unable to replicate the TORCH placebo comparison findings, this would be one of the aspects that we would investigate as a possible reason, and would be an interesting finding to report. We would also plan to perform a new analysis where we compare FP/SAL to Salmeterol, one of the other comparator groups from TORCH to see if we can replicate results when a comparator drug has been used.

#	Feedback	Response and details of any changes made
		In relation to this point, we have updated the Exposures, outcomes and co-variates section (Exposures subheading) to include the following additional sentence at the end of the second paragraph: “If the results for our FP/SAL vs no treatment comparison are not consistent with the TORCH FP/SAL vs placebo results (see Statistical analysis – Validation of results against TORCH section for a definition of consistent), we will perform additional analyses where instead of using a no-treatment comparator group, our Objective 1 comparator group will be people exposed to Salmeterol, one of the other comparator groups from the TORCH trial.”. We also noticed that the list of other treatments for Objective 1 following this paragraph had an “(f)” but no “(e)” so have updated this. We have also added the following sentence to the end of the Statistical analysis, Validation of results against TORCH section: “If we go on to compare FP/SAL with Salmeterol alone (see Exposures, outcomes and co-variates section, Exposures subheading), the 95% confidence interval would also need to exclude 1, and the rate ratio would need to be between 0.81 and 0.95 (compared with the TORCH FP/SAL versus Salmeterol result of 0.88, 95% CI 0.81-0.95).
4	In TORCH, approx. 60% got medication prior to run in, that is 22% ICS, 8% LABA and 30% ICS/LABA – 40% received no medication. This is nice information when looking at the results. Ideally, they	We are already planning to match on this characteristic. Please see the list of TORCH baseline characteristics that we will be matching on provided in the section Selection of participants, Objective 1,

#	Feedback	Response and details of any changes made
	should be matched on this (but it might be close to impossible) – however, if they all receive ICS/LABA it can be because they are sicker than in TORCH (and not just randomized be chance) which might influence the result (the doctor chose the medication for a reason)?	Step 3 (which includes previous treatment with inhaled corticosteroids or long acting beta-agonists). Therefore, no additional changes to the manuscript have been made in relation to this point.
5	How will you evaluate the effects, are they all equally important and how close to TORCH do they have to be to be called the same – could you elaborate?	We already include a dedicated section entitled Validation of results against TORCH (in the Statistical analysis section) that defines the two key criteria that must be met for us to conclude that the results are consistent (specifying the acceptable rate ratio range, and that the 95% confidence intervals must not cross 1). Therefore, no additional changes to the manuscript have been made in relation to this point.
6	I think it would be wise and fair to track how many people that change medication during the study (it is a lot I can assure you) – for instance compare beginning of the 3 years with the end – especially if your results differ from TORCH this might be the most obvious reason (as well as the once described above).	We already plan to include time to treatment change as one of our study outcomes. This is mentioned in the Abstract - Methods and analysis section (penultimate sentence), the Aims and objectives section (first sentence) and the Exposures, outcomes and co-variates section (Outcomes subheading, point 4.). When considering this point, we noted that a description of how we are going to analyse time to treatment change had been omitted from the Statistical analysis, Methods of analysis section, and therefore updated the 4th sentence of this section so that it reads as follows: “Time to mortality, first pneumonia and treatment change will be analysed using Cox proportional hazards regression.”
Objective 2		
7	As I read it – again you only look at Seretide in the exposed group and no Seretide in the unexposed group?! – this time in a larger group (also including oxygen, comorbidities etc.) – or what (or is it only in oxygen, comorbidities	For Objectives 2 and 3, you are correct that the comparison is FP/SAL treated vs FP/SAL untreated. Furthermore, the groups that we analyse for Objective 1 and Objective 2 will be completely

#	Feedback	Response and details of any changes made
	etc.)? You write additional criteria but is it additional to the diagnosis (I think so) or to the criteria mentioned in objective 1 – please specify as you write about objective 1 in objective 2, so this is unclear at the moment.	different to the Objective 1 group i.e. the criteria for inclusion into Objectives 2 and Objectives 3 differ from each other and from Objective 1, so different populations will be analysed within each Objective. We have made the following changes to the manuscript in order to clarify these points: Exposures, outcomes and co-variates section, Exposures subheading, 3rd paragraph and list for Objectives 2 and 3: Added “(all versus no treatment)” Changed the first list entry so that it reads “FP/SAL” (rather than “No treatment”). Selection of participants, Objective 2 subheading, the first sentence of the first paragraph now reads: “We will select separate cohorts of patients who have a valid COPD diagnosis in the CPRD⁸ and who would not have been eligible for inclusion in the TORCH trial (or our Objective 1) due to the following characteristics: (1) age >80 years OR (2) history of lung surgery OR (3) history of long term oxygen therapy OR (4) evidence of drug/alcohol abuse OR (4) an asthma diagnosis at any time prior to inclusion OR (5) substantial comorbidity.” And the first sentence of the second paragraph now reads: “Participants for each of the Objective 2 cohorts will be selected in a similar fashion to the Objective 1

#	Feedback	Response and details of any changes made
		cohort, with the amended eligibility criteria specified above applied (i.e. Step 1 will be modified for selection of each of the Objective 2 cohorts).” Selection of participants, Objective 3 subheading, the first sentence of the first paragraph now reads: “We will select separate cohorts of patients who have a valid COPD diagnosis in the CPRD⁸ and who would not have been eligible for inclusion in the TORCH trial (and therefore also not eligible for our Objective 1) due to the following characteristics: (1) >60% predicted FEV1 (or >50% plus MRC breathlessness scale 1 or 2, or >50% plus COPD Assessment Test (CAT) score <10) AND (2) no exacerbations in the year post COPD diagnosis.”
8	As I see it you have the same medication issues here as in objective 1.	Please see the responses (1) and (6) above, which all also apply to Objective 2 (given that steps 1-2 and 4-6 of Objective 1 also relate to Objective 2, as specified in the Selection of participants, Objective 2 subheading).
9	I suppose you will look at the same outcome parameters?	Yes, we plan to analyse the same outcomes for every Objective. In order to clarify this point, we have amended the first sentence of the Statistical analysis - Methods of analysis section so that it now reads: “For all objectives, comparisons will be made according to FP/SAL (or other drugs being analysed as specified in the Exposure section) status for rate of COPD exacerbation, pneumonia and mortality over 3 years.”
Objective 3		

#	Feedback	Response and details of any changes made
10	See objective 2. Identical questions.	See above, responses to Objective 2 queries above also include relevant changes to Objective 3.
Exposures		
11	You write about secondary analysis. I am not sure what you mean/how you will do this with respect to medication – this is baseline information I suppose? Why is no medication only present under objective 2 and 3 (the untreated in objective 1 are not truly untreated or?)	For the secondary analyses looking at drug exposures other than FP/SAL, our aim is to apply the same methods that we used for Objective 1 (comparing FP/SAL with no FP/SAL), but specific to the new medication under study (e.g. LAMA vs no LAMA). We will therefore apply Steps 1-2 and 4-6 from the Selection of participants – Objective 1 section. Step 3 will not be required, because while we want to select our exposed and unexposed participants as though we were performing a trial, we don't need to match to TORCH (Step 3) as we are not aiming to create a TORCH-analogous cohort to allow comparison of results with TORCH. In order to clarify this point, we have updated the 2nd and 3rd sentences of the 3rd paragraph of the Exposures section to read: “Selection of unexposed and exposed people for each of these drugs will follow Steps 1 – 6 detailed above in the Selection of Participants –Objective 1 section, although Step 3 will be omitted (as these cohorts will not need to be matched to TORCH). The other treatments we plan to include are as follows:” Please also see the change made to the Method of analysis section referred to in (9) above. This was an updating error, and has been corrected (see response to (7) above).

#	Feedback	Response and details of any changes made
12	You write under handling measurement of adherence to medication, that you have performed a sample on 30 patients over 1 year and found 2/3 were still receiving Seretide. Try to repeat that for 2 and 3 years – you will indeed be surprised to see how dramatically that number drops. I have done a lot of database research with medication and it is NOT easy! 1 year studies are ok but the longer the period – the more disappointing and confusing the results.	We appreciate this important point related to adherence. The final sentence of the Handling measurement of adherence to medication section states that in the event that Objective 1 detects a null or poorer treatment effect than anticipated (rate ratio > 0.9), we will conduct a sensitivity analysis restricted to people estimated to be covered by FP/SAL treatment for 80% of their follow up. We believe that this approach will allow us to assess the impact of adherence being poorer in routine care than in the TORCH trial. We will also assess and report on the level of adherence beyond 1 year in our selected cohort for Objective 1, and have added the following new sentence to this section in order to clarify this: “We plan to assess adherence for the cohort that we select for Objective 1 beyond 1 year and report the findings.”
Author-driven changes to this version		

#	Feedback	Response and details of any changes made
1	Clarifications to wording throughout the text	Abstract – Introduction “analysing” changed to “to the analysis of” Introduction – Background and rationale 3rd paragraph “in this setting” added. 4th paragraph “analysing” replaced by “to the analysis of” Aims and objectives 1st paragraph “and COPD” changed to “and a number of COPD outcomes including” Selection of participants Objective 1 – Step 6 Deleted “The index date will be the start of follow-up.” Exposures, outcomes and co-variates Title changed to Exposures, outcomes and covariates Paragraph 2: “incident” deleted (as people may have already been prescribed FP/SAL during a previous eligibility period)

#	Feedback	Response and details of any changes made
		Statistical analysis Propensity score for addressing confounding 2nd sentence, added “(as listed in the covariates section above)” Final sentence of 1st paragraph changed to: “We have substantial prior experience of building propensity models¹³⁻¹⁶” 2nd paragraph removed (as already been covered in Selection of participants, Objective 1 section) Final paragraph: “variable list used for the” added Methods of analysis “” added around intention-to-treat Validation of results against TORCH 2nd sentence, “intervals” replaced with “limits”

VERSION 3 – REVIEW

REVIEWER	Anders Løkke Department of Respiratory Diseases and Allergy Aarhus University Hospital Noerrebrogade 44 8000 Aarhus C Denmark
REVIEW RETURNED	27-Jan-2018
GENERAL COMMENTS	I have no further comments. You have made some fine changes in the manuscript. I congratulate you with this nice and advanced protocol. Best of luck.